# Preparation and Characterization of a One-Step Electrospun Poly(Lactic Acid)/Wormwood Oil Antibacterial Nanofiber Membrane

**DOI:** 10.3390/polym15173585

**Published:** 2023-08-29

**Authors:** Xiaoyan Tang, Xun Guo, Yongchao Duo, Xiaoming Qian

**Affiliations:** School of Textile Science and Engineering, Tiangong University, Tianjin 300387, China; xiaoyantang123@163.com (X.T.);

**Keywords:** electrospinning, nanofiber, poly(lactic acid), wormwood oil, natural antimicrobial

## Abstract

With the continuous improvement of the standard of living, people are increasingly inclined towards natural, green, and environmentally friendly products. Plant-based products that are safe, natural, non-toxic, and beneficial to human health are often more favored. Poly(lactic acid) (PLA) is a polymer obtained through lactate polymerization using renewable plant resources such as corn and has excellent biocompatibility and biodegradability. It is widely used in the field of food packaging. Wormwood oil (WO) is an oil extracted from the stems and leaves of Artemisia plants, and it has broad-spectrum antibacterial properties. In this article, through electrospinning technology, wormwood oil was directly incorporated into PLA, giving the PLA nanofiber membrane antioxidant and antibacterial functions. Various parameters such as voltage (11 KV, 13 KV, 15 KV), spinning solution concentration (8%, 10%, 12%), distance (15 cm, 17 cm, 19 cm), and feeding rate (0.4 mL/h, 0.5 mL/h, 0.6 mL/h) were explored, and the resulting spun fibers were characterized. Through SEM characterization, it was found that when the spinning voltage was 13 KV, the spinning solution concentration was 10%, the distance was 17 cm, and the feeding rate was 0.5 mL/h, the nanofiber membrane had a smooth morphology without bead formation, with an average diameter of 260 nm. The nanofiber membrane was characterized using FTIR, TG, and SEM, confirming the successful incorporation of artemisia essential oil into PLA. The prepared antimicrobial nanofilm was subjected to antimicrobial testing, and the results showed that as the concentration of the essential oil increased, the inhibition zones also increased. When wormwood oil concentration was 4%, the diameter of the inhibition zone for *Staphylococcus aureus* increased from 1.0 mm to 3.5 mm, while the diameter of the inhibition zone for *Escherichia coli* increased from 2.0 mm to 4.5 mm.

## 1. Introduction

Antimicrobial nanofilm is a novel type of antimicrobial material that exhibits unique nano-effects and long-lasting antimicrobial properties, effectively inhibiting the growth and reproduction of microorganisms and reducing potential safety risks [1,2]. There are two forms of nanofilm: one involves incorporating antimicrobial nanoparticles into polymer materials to confer antimicrobial effects to the polymers, while the other involves processing antimicrobial materials at nanoscale to create nanosized antimicrobial materials. Common nano antibacterial materials involve incorporating nanoparticles such as silver (Ag) or gold (Au) ions into polymers. These materials exhibit long-lasting and effective antibacterial properties. However, the use of metallic nanoparticles poses potential risks to human health and the environment. When the concentration exceeds a certain proportion, direct harm to the human body may occur. With further research on nanomaterials, a new trend has emerged in the development of nano antibacterial materials that consist of nanofibers and natural antibacterial substances.

Plant essential oils, also known as aromatic oils, are highly concentrated natural bioactive substances obtained from the leaves, stems, or bark of plants through distillation or extraction methods. They are often referred to as “liquid gold” due to their valuable properties. Essential oils are liquid at room temperature, insoluble in water, and soluble in most organic solvents. The composition of plant essential oils is diverse and complex, generally containing terpenoids [3,4], aromatic compounds [5], and fatty compounds, among others. Plant essential oils have various characteristics, including air purification, cellular nourishment, mind–body balance, and natural preservation. Wormwood, also known as moxa, is an herbaceous plant of the Asteraceae family. In ancient China, it was burned to repel mosquitoes and prevent diseases [6]. Over time, various wormwood products have been developed, including wormwood oil, moxibustion patches, wormwood tea, wormwood pillows, health belts, and mosquito coils. Wormwood oil (WO) is extracted from the stems and leaves of wormwood and is obtained through methods such as distillation, supercritical extraction, organic solvent extraction, and enzyme extraction [7]. Wormwood oil is a green and transparent oily liquid with volatility. Chromatographic analysis of wormwood volatile oil indicates that monoterpenes, mainly represented by terpinenes, dominate the composition [8]. This is also confirmed by the findings of Martin et al. [9], who identified Z-cyclohexene oxide, artemisia alcohol, and artemisia esters as the main components found in supercritical extracts and steam-distilled essential oils of wormwood. Wormwood oil has a distinctive odor. Research suggests that the odor is caused by bitter compounds present in the plant extract, such as sesquiterpene lactone and sesquiterpene lactone dimer [10]. β-caryophyllene, one of the components in WO, has been shown by Gonzalez et al. [11] to enhance immune function and promote cell growth at wound sites. Zeng et al. [12] found that the sesquiterpene dimer artemisinin D in WO exhibits significant anti-neuroinflammatory effects, while Artemisia extracts demonstrate broad-spectrum antibacterial and antifungal activities. Moreover, Mamatova et al. [13] identified artemisinin, rutin, isorhamnetin, and scopolamine as components in Artemisia, which can be used for the development of effective antibacterial and antifungal drugs, as well as supporting treatments for infectious diseases. The mechanisms of essential oils on microorganisms are complex. It is generally believed that the antimicrobial effects of essential oils rely on their hydrophilic or lipophilic nature. Terpenoids, being lipophilic substances, can influence the activity of membrane-bound enzymes, such as those involved in respiration pathways [14]. Some lipophilic components of essential oils can act as uncouplers, interfering with proton transport on membranous vesicles and disrupting adenosine diphosphate (ADP) phosphorylation. Specific functional groups present in terpenoids, such as aldehydes and alcohols, can interfere with membrane integrity or the synthesis of related enzyme proteins, thus inhibiting microbial growth or activity [15].

PLA is a biodegradable material that possesses excellent mechanical and physical properties, as well as high tensile strength and elongation [16]. PLA is suitable for various processing methods, such as blow molding and thermoforming. It is easy to process and has a wide range of applications. It can be used to manufacture various plastic products, food packaging, disposable food containers, non-woven fabrics, industrial and domestic textiles. It can also be further processed into agricultural fabrics, healthcare textiles, cleaning cloths, sanitary products, outdoor UV-resistant fabrics, tent fabrics, and carpet surfaces, among others, making it a highly promising market. PLA exhibits good compatibility and biodegradability [17]. It is extensively used in the medical field for producing disposable medical equipment, absorbable surgical sutures, and low molecular weight PLA serves as a drug delivery packaging material. PLA is made from renewable plant resources such as starch, making it cost-effective and widely available. As the final decomposition products of PLA can be naturally absorbed and utilized by the environment, it adheres to the principles of a circular economy [18]. Moreover, PLA reduces dependency on non-renewable resources like petroleum and aligns with the requirements of sustainable development. PLA films have excellent breathability, oxygen permeability, and carbon dioxide permeability, as well as odor barrier properties. Biodegradable plastics are known to attract viruses and molds, raising concerns about safety and hygiene. However, PLA, being a biodegradable plastic, exhibits anti-mold characteristics [19]. When incinerated, PLA has the same calorific value as burning paper, half that of traditional plastics like polyethylene. Moreover, incineration of PLA does not release toxic gases such as nitrogen compounds or sulfides. The human body also contains lactic acid in its monomer form, indicating the safety of such biodegradable products. Kayaci et al. [20] incorporated a complex of triclosan (TR) and beta-cyclodextrin (β-CD) into a PLA solution and successfully prepared PLA/TR/β-CD nanofiber membranes using the electrospinning technique. The results indicated that due to the inclusion complexation of β-CD with TR, the degradation of TR in the environment was reduced, resulting in significantly larger inhibition zone diameters of the PLA/TR/β-CD nanofiber membrane against *Escherichia coli* (*E. coli*) and *Staphylococcus aureus* (*S. aureus*) compared to the PLA/TR nanofiber membrane (3.7 cm vs. 3.0 cm; 3.50 cm vs. 2.83 cm). Ramos et al. [21] prepared a nanocomposite film by incorporating resveratrol and silver nanoparticles into PLA. The addition of resveratrol and silver nanoparticles improved the thermal, optical, and barrier properties of PLA. The prepared nanocomposite film exhibited 90% degradation after 14 days under composting conditions. Wei Jing [22] fabricated antibacterial nanofiber membranes using PLA and chitosan nanowhiskers (CNW) via electrospinning and applied them for strawberry packaging. It was found that the composite film containing 7% CNW exhibited significant antibacterial activity against *E. coli* and *S. aureus*, with inhibition rates of 94.5% and 91.7%, respectively. Qin [23] used electrospinning technology to prepare PAN/Cur@OD composite fiber membranes. These composite membranes have excellent antibacterial properties and have potential application value in areas such as textile fabrics, food preservation films, and biomedicine.

This study aimed to impart antibacterial properties to a polylactic acid (PLA) polymer by incorporating wormwood essential oil using a one-step electrospinning method. By optimizing the electrospinning conditions, nanocomposite materials with smooth surfaces, an average diameter of 260.5 nm, and excellent antibacterial properties were successfully prepared. The nanocomposite was characterized using techniques such as scanning electron microscopy (SEM), transmission electron microscopy (TEM), Fourier-transform infrared spectroscopy (FTIR), and differential scanning calorimetry (DSC), which confirmed the successful composite formation of wormwood essential oil with a PLA polymer. Antibacterial tests demonstrated that the prepared nanocomposite antimicrobial film exhibited effective antibacterial properties. As the WO content increases, the antibacterial zones of the nanofiber membrane against Escherichia coli and Staphylococcus aureus also increase.

## 2. Materials and Methods

### 2.1. Materials

PLA (average MW = 80,000) and Hexafluoroisopropanol (analytical grade, purity > 98%) were purchased from Shanghai Aladdin Bio-Chem Technology Co., Ltd. (Shanghai, China). Wormwood oil (analytical grade, purity > 98%) was purchased from China Merchants Health Industry (Herb) Co., Ltd. (Huanggang, China). Agar powder and peptone were purchased from Hangzhou Baisi Biotechnology Co., Ltd. (Hangzhou, China). Yeast infusion powder was purchased from Macklin Biochemical Technology Co., Ltd. (Shanghai, China) The *E. coli* (CMCC44103) was purchased from Shanghai Hu zheng Biotechnology Co., Ltd., (Shanghai, China) and the *S. aureus* (CMCC26003) used in this experiment was purchased from Shanghai Xuan ya Biotechnology Co., Ltd. (Shanghai, China).

### 2.2. Characterization

The viscosity of the spinning solution was measured at 25 °C using a digital viscometer (NDJ-8S). The morphology of the nanofiber membrane was observed using a desktop scanning electron microscope (SEM, Phenom XL, Phenom-World, Eindhoven, The Netherlands) at an acceleration voltage of 15 kV. The nanofiber membrane was then coated with platinum (Pt) using an ion sputter coater for 120 s. The composition of PLA, wormwood essential oil, and the antibacterial nanofiber membrane was analyzed using a Fourier transform infrared spectrometer (FTIR, Nicolet iS50, Thermo Fisher Scientific, Shanghai, China) to confirm the successful composite formation of wormwood essential oil with PLA. The thermal stability of the antibacterial nanofiber membrane material was tested using a thermogravimetric analyzer (TG, 209 F3 Tarsus, Zetzsch, German). The antibacterial nanofiber membrane was analyzed using a transmission electron microscope (TEM, Hitachi H7650, Hitachi, Japan).

### 2.3. Antibacterial Assays

The *S. aureus* and *E. coli* bacterial strains were activated and selected from well-grown single colonies for subculturing. After two passages, the bacterial strains were inoculated into nutrient broth and incubated at 37 °C for 18 h. Then, a serial dilution was performed using sterile nutrient broth. Subsequently, nanofiber membranes (PLA nanofiber membrane and PLA nanofiber membrane with different concentrations of essential oil) were punched into uniform circular discs with diameters of 2 cm using a hole puncher. The discs were then placed under a UV lamp for 2 h to sterilize them. Sterile nutrient agar plates were prepared, and 0.1 mL of the bacterial suspensions mentioned earlier was evenly spread onto each plate to obtain plates with bacteria. The sterilized antimicrobial membranes were placed on the agar plates with bacteria and incubated inverted at 37 °C for 24 h. The inhibition zones were observed, and their sizes were recorded.

### 2.4. Preparation of Antimicrobial Nanofilms

A certain amount (1~3 g) of PLA pellets were weighed and dissolved in hexafluoroisopropanol solvent. The mixture was stirred at room temperature for 2 h to obtain PLA spinning solutions with mass fractions of 8%, 10%, and 12%. The morphology of the nanofiber membranes obtained from electrospinning was observed using a scanning electron microscope (SEM). Based on the SEM images, an appropriate PLA spinning solution was selected and solutions with wormwood essential oil concentrations of 2%, 4%, and 6% were prepared. The solutions were then subjected to electrospinning at a temperature of 20 ± 2 °C and a relative humidity of 40 ± 2%, and the morphology of the resulting nanofiber membranes was observed using an SEM. Under the optimized electrospinning solution composition mentioned above, process parameters were adjusted to obtain nanofiber membranes with good morphology and uniform diameter distribution. The specific operation procedure involved injecting the solution stored in a 10 mL syringe through a size 20 stainless steel needle, with the injection pump connected to the positive electrode of the high-voltage power supply and the receiving plate connected to the negative electrode. The spinning process was initiated by applying power, which caused the solution at the needle tip to be stretched and elongated, forming electrospun nanofiber membranes on an aluminum foil receiving plate. The investigated process parameters included a voltage of 11~15 kV, an injection speed of 0.4~0.6 mL/h, a distance of 20 cm between the receiving plate and needle tip, and a duration of 4 h. Finally, the obtained nanofiber membranes were dried at 35 °C for 12 h for morphology observation or other structural characterizations (at a temperature of 20 ± 2 °C and a relative humidity of 40 ± 2%).

## 3. Results and Discussion

### 3.1. Scanning Electron Microscope (SEM) Analysis

The solution and process parameters are the main factors that affect the morphology of fibers [24,25]. Excessively high and low viscosity of the solution hinders the stretching of the fibers, leading to poor uniformity. In the electrospinning process, the concentration of the polymer mainly affects the spinnability and the diameter of the fibers. Generally, a higher concentration of the spinning solution results in larger fiber diameter [26,27,28]. Currently, when using PLA for electrospinning, the concentration typically ranges from 8% to 14%. Studies have found that when the mass concentration of PLA is 14%, the viscosity of the spinning solution is too high, leading to clogging of the needle holes and preventing continuous nanofibers. Therefore, experiments were conducted to examine the influence of PLA concentrations of 8%, 10%, and 12% on the morphology of the resulting nanofiber films. From Figure 1, it can be observed that when the PLA concentration is 12%, the diameter distribution of the resulting fibers is uneven, and the fibers tend to stick to each other. This is because the viscosity of the 12% PLA solution is significantly higher than that of the 10% PLA solution (0.432 Pa·S vs. 0.361 Pa·S, Table 1), which hinders the stretching and differentiation of the fibers. When the PLA concentration is 8%, the resulting fibers exhibit a beading phenomenon. This is because the solubility of the solution is low, resulting in weak intermolecular adhesion and insufficient stretching of the fibers [29]. Therefore, the PLA concentration of 10% (*w*/*w*) was chosen, which resulted in an average fiber diameter of 260 nm.

Generally, the spinning voltage affects the shape of the droplets at the needle tip and the flight time of the droplets. When the spinning voltage is relatively low, the spinning solution cannot be effectively stretched, resulting in a larger fiber diameter. Figure 2 shows that as the spinning voltage increases to 15 kV, the solution droplets are well-stretched under the action of the electric field, resulting in a uniform fiber diameter distribution without beading. Increasing the voltage also reduces the solvent evaporation time. Therefore, when the voltage is increased to 15 kV, the strong electric field force minimizes the solvent evaporation, resulting in an uneven fiber diameter distribution and droplet shape on the collector. Therefore, 15 kV is chosen as the appropriate spinning voltage.

The collector distance primarily affects the flight time of the droplets and the efficiency of solvent evaporation. From Figure 3, it can be observed that when the collector distance increases from 15 cm to 17 cm, the average diameter of the fibers decreases, and the fiber diameter distribution becomes more uniform. This is because increasing the collector distance prolongs the flight time of the droplets, allowing for sufficient stretching of the fibers. However, when the collector distance is set to 19 cm, the non-uniformity of the fiber diameter distribution increases. This is because increasing the collector distance weakens the electric field strength, resulting in a decrease in the surface charge per unit area at the needle tip. As a result, the electric field force acting on the charged droplets is insufficient to overcome the surface tension, leading to increased instability in the fiber distribution. This finding is consistent with the research by Ki et al. [30]. Therefore, we chose 17 cm as the appropriate collector distance.

The injection rate affects the volume of the spinning solution. A higher injection rate results in a larger volume of spinning solution accumulated at the needle tip. With a constant electric field intensity and collector distance, this can impact the quality of the Taylor cone formed by the injection needle. It is easier for a spray jet to form smaller droplets, and it requires a longer time for the solvent to evaporate, leading to droplet aggregation and an increase in fiber diameter. From Figure 4, it can be observed that when the injection rate increases from 0.4 mL/h to 0.6 mL/h, the same phenomenon occurs as described above. When the injection rate is 0.5 mL/h, the nanofiber film obtained has a uniform diameter distribution and a smooth surface. Therefore, we choose an injection rate of 0.5 mL/h as the appropriate injection rate.

Therefore, for a solution composition of 10% (*w*/*w*) PLA/4% (*w*/*w*) wormwood oil, the optimized process parameters are a spinning solution concentration of 10% (*w*/*w*), spinning voltage of 13 kV, injection rate of 0.5 mL/h, and collector distance of 17 cm. Under these conditions, the obtained nanofiber film exhibits a good morphology without spindle-shaped beads, with an average diameter of 254 nm, as shown in Figure 5.

### 3.2. Fourier Transform Infrared Spectra (FTIR) Analysis

To determine the successful fabrication of the antibacterial nanofiber membrane, we conducted an infrared spectroscopy test. Infrared spectra scans were performed on the PLA nanofiber membrane, wormwood oil, and the antibacterial nanofiber membrane.

Figure 6 demonstrates that the characteristic absorption peaks of PLA are observed at 1756 cm^−1^ (C=O stretching vibration) and 1438 cm^−1^ (C-H stretching vibration) in the FTIR spectrum of the PLA nanofiber membrane. The FTIR spectrum of the Wormwood oil shows absorption peaks at 1410 cm^−1^ (O-H stretching vibration) and 2958 cm^−1^ (C-H stretching vibration), which correspond to the major active components, camphor-4 and α-terpineol, of the wormwood oil [31].

In the infrared spectrum of the PLA/WO nanofiber membrane, all major characteristic peaks of both PLA and WO can be clearly observed. This indicates the successful fabrication of the antibacterial nanofiber membrane. The main peaks obtained at 1760 cm^−1^ and 1440 cm^−1^ correspond to the C=O stretching vibration and C-H stretching vibration of PLA, respectively. The peaks at 1413 cm^−1^ and 2960 cm^−1^ represent the O-H stretching vibration and C-H stretching vibration, respectively. It is worth noting that the FTIR peaks of the antibacterial nanofiber membrane show slight shifts compared to those of the individual components. This suggests a certain interaction between the PLA and WO components in the blend, further confirming that no chemical reaction occurred between the two components.

### 3.3. Thermogravimetric Analyzer (TG) Analysis

Figure 7 shows the thermogravimetric (TG) and derivative thermogravimetric (DTG) curves of PLA, WO, and PLA/WO materials heated from 30 °C to 550 °C at a heating rate of 10 °C/min. From the graph, it can be observed that all three samples exhibit relatively low mass loss in the initial stage, which is attributed to the presence of a small amount of moisture in the samples. In the TG and DTG curves of PLA, the mass change of PLA begins at 313 °C, with the peak weight loss rate occurring at 339 °C. For WO, the mass change starts at 107 °C, and the maximum weight loss rate is observed at 150 °C. In the case of the PLA/WO curve, the mass loss occurs in two stages. The first stage of weight loss begins at 111 °C, with the maximum weight loss rate observed at 148 °C. The second stage of weight loss starts at 327 °C, and the peak weight loss rate is observed at 350 °C.

The two weight loss stages in the TG and DTG curves of PLA/WO correspond to the weight loss of PLA and WO, respectively. This further confirms the successful fabrication of the antibacterial nanofiber membrane by blending PLA and WO.

### 3.4. Transmission Electron Microscope (TEM) Analysis

TEM analysis was conducted to observe the morphology of the antibacterial nano-fiber membrane at high resolution. From Figure 8, it can be observed that the antibacterial nanofiber membrane consists of two components. The darker component is WO, while the lighter component is PLA. WO is mixed with PLA, and its dispersion in PLA is random.

### 3.5. Antibacterial Performance and Analysis

The antibacterial performance of the antibacterial nanofiber membrane prepared under the optimized conditions was evaluated using the inhibition zone method, examining the inhibitory effect against *S. aureus* and *E. coli*. Through three repeated experiments, it was found that the nanofiber membrane exhibited a certain inhibitory effect against both *S. aureus* and *E. coli*. The PLA nanofiber membrane showed poorer antibacterial performance against *S. aureus* and *E. coli*, and as the WO content increased, the inhibition zones continuously increased. It can be observed from Figure 9 that the increase in essential oil content consistently improved the antibacterial effect against *S. aureus* and *E. coli*. Table 2 shows the inhibition zones of PLA nanofiber membranes with different contents of WO against *E coli* and *S. aureus*. It can be observed that the inhibition zone sizes of pure PLA against *S. aureus* and *E. coli* are 1.0 mm and 2.0 mm, respectively. When the WO content is 4%, the nanofiber membrane shows inhibition zones of 3.5 mm and 4.5 mm against *S. aureus* and *E. coli*, respectively. Increasing the WO content enhances the antimicrobial effectiveness of the nanofiber membrane.

## 4. Conclusions

In conclusion, we successfully prepared antibacterial nanofiber membranes of PLA loaded with natural antibacterial material wormwood oil using the electrospinning method. Through process optimization, the nanofiber membranes prepared with a solution concentration of 10% (*w*/*w*), voltage of 13 kV, feed rate of 0.5 mL/h, and collector distance of 17 cm exhibited favorable morphology with uniform diameter distribution and an average diameter of 260 nm.

Characterization was conducted through FTIR, TG, and TEM. FTIR results showed that the prepared antibacterial nanofiber membrane exhibited all the main peaks of PLA and WO. TG curves indicated two weight loss stages, corresponding to the weight loss of PLA and WO. The first stage corresponds to the weight loss of WO, while the second stage corresponds to the weight loss of PLA. From TEM, it is clear that the prepared nanofiber membrane consists of two components, with one component being WO and the other component being PLA.

From the antimicrobial test, it can be observed that pure PLA has almost no antibacterial effect on *E. coli* and *S. aureus*. With the increase in wormwood essential oil content, the antibacterial zones against *S. aureus* and *E. coli* also increased, with a greater inhibitory effect on *E. coli*. These characterizations indicate the successful combination of wormwood oil and PLA, demonstrating a certain level of antibacterial activity. PLA, known for its excellent biocompatibility and mechanical properties, is utilized in the production of antibacterial nanofiber membranes that exhibit certain antimicrobial properties. Such membranes hold significant potential in the manufacturing of medical and hygiene products, such as masks and dressings.

## Figures and Tables

**Figure 1 polymers-15-03585-f001:**
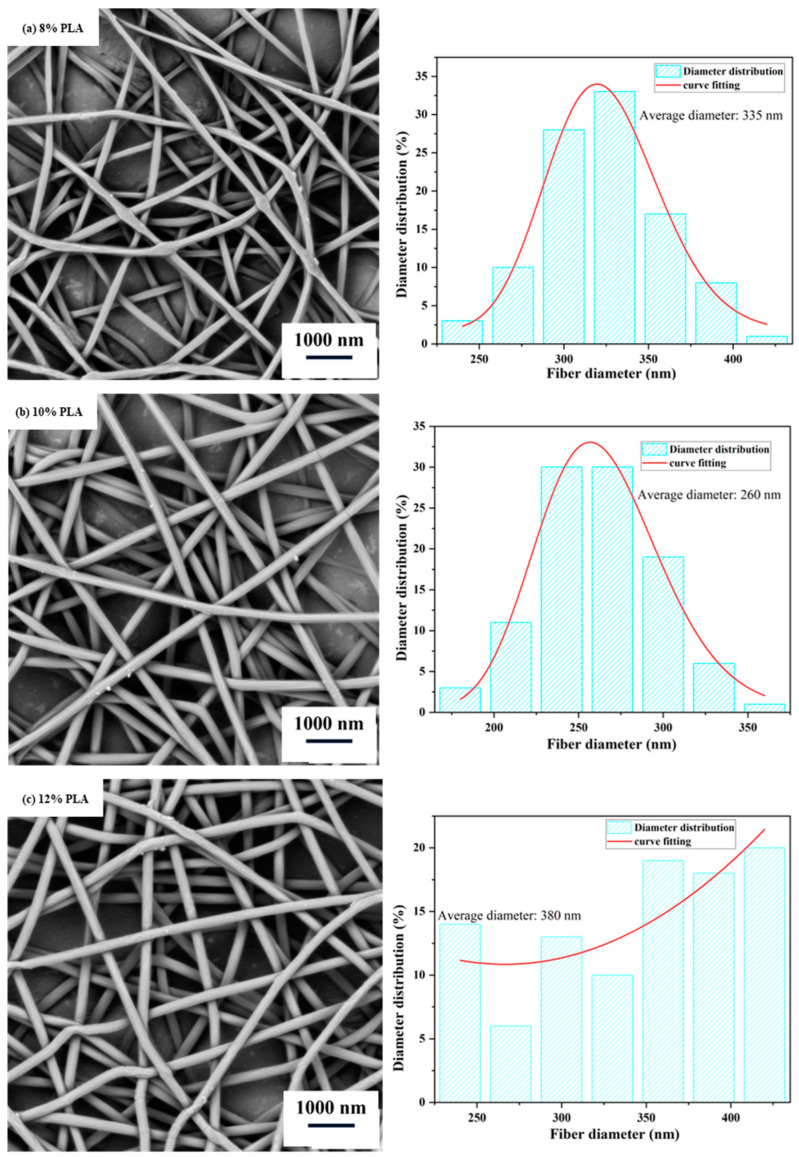
Scanning electron microscope (SEM) images and diameter distribution of electrospun nanofiber films at different PLA concentrations. (Spinning process parameters: Injection rate of 0.5 mL/h, spinning voltage of 13 kV, and collector distance of 17 cm. The diagram on the right depicts the distribution of nanofiber diameters.).

**Figure 2 polymers-15-03585-f002:**
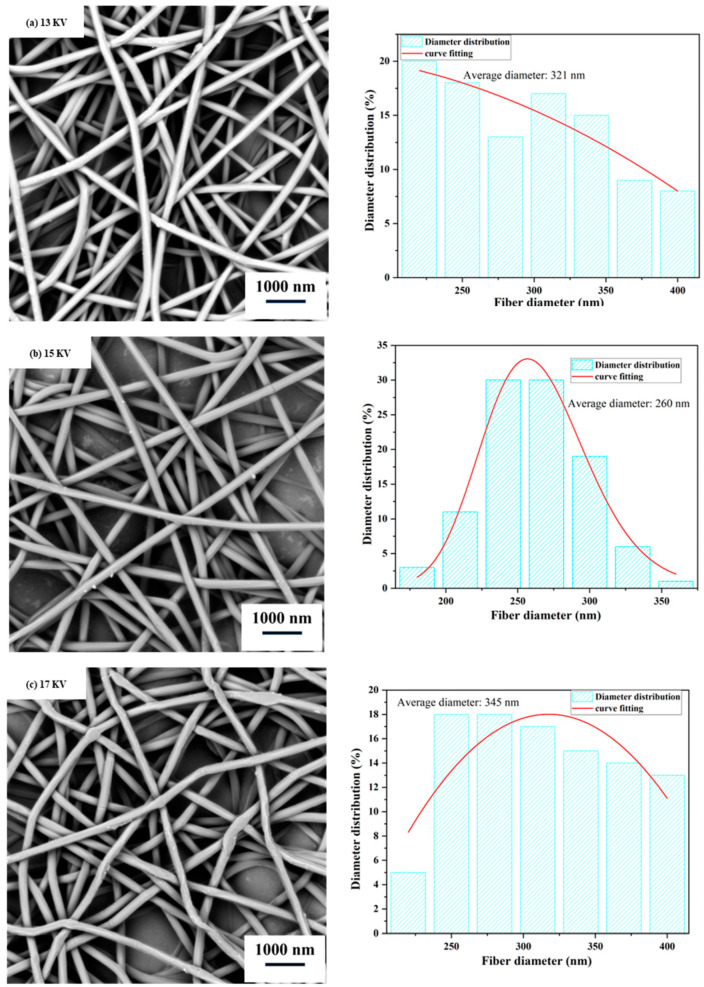
Scanning electron microscope (SEM) images and diameter distribution of nanofiber films at different spinning voltages. (Spinning process parameters: Injection rate of 0.5 mL/h, solution concentration of 10% (*w*/*w*), and collector distance of 17 cm. The diagram on the right depicts the distribution of nanofiber diameters).

**Figure 3 polymers-15-03585-f003:**
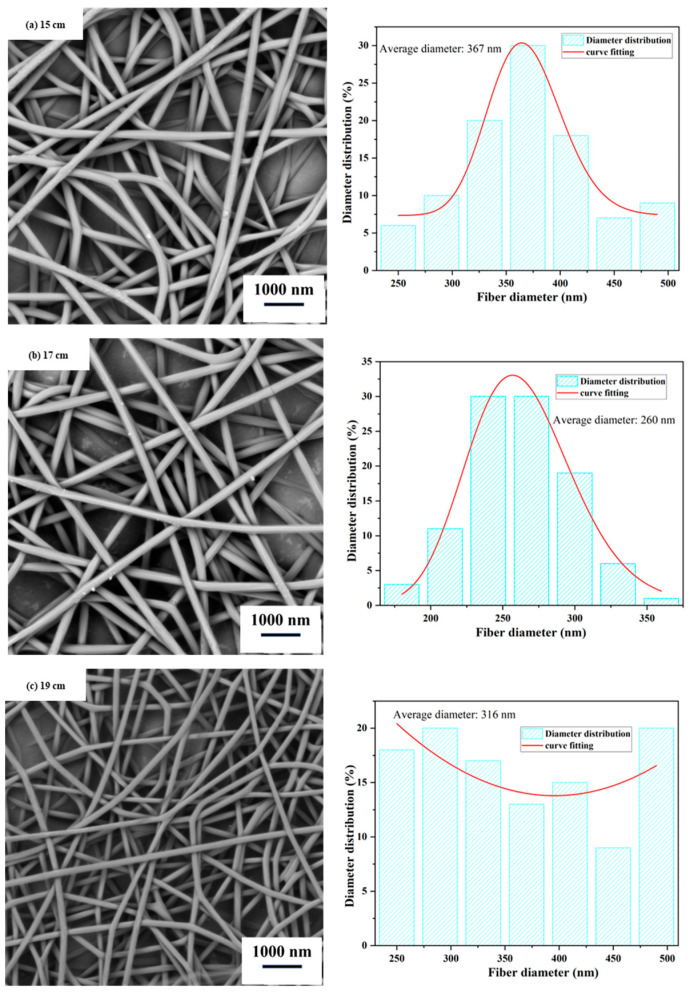
Scanning electron microscope (SEM) images and diameter distribution of nanofiber films at different collector distances. (Spinning process parameters: Injection rate of 0.5 mL/h, solution concentration of 10% (*w*/*w*), and spinning voltage of 13 kV. The diagram on the right depicts the distribution of nanofiber diameters.).

**Figure 4 polymers-15-03585-f004:**
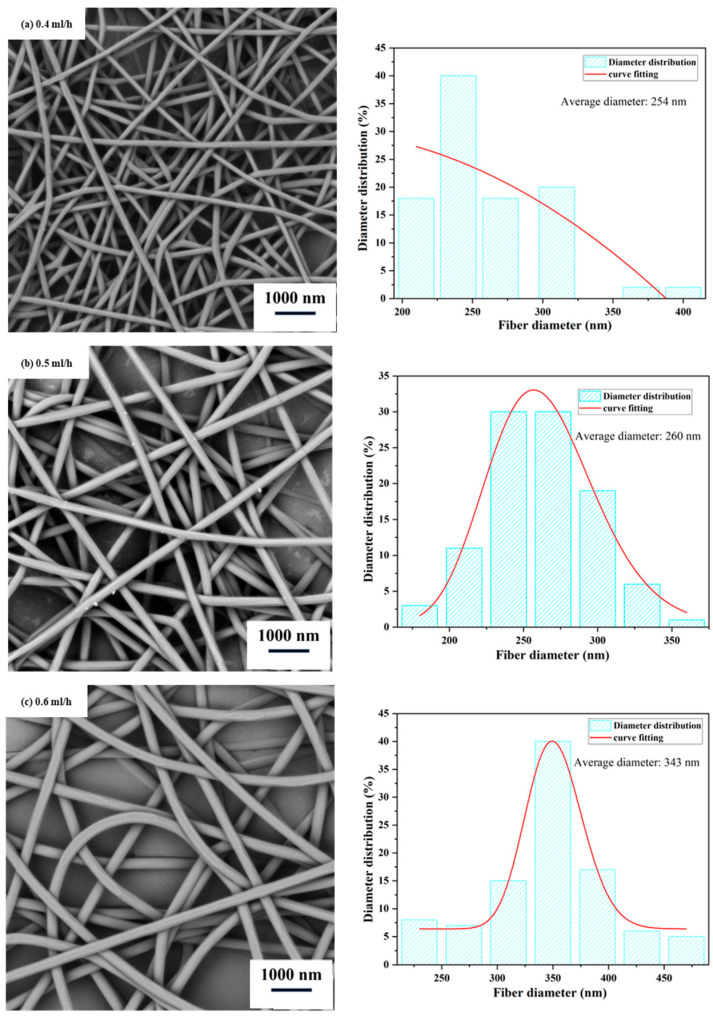
Scanning electron microscope (SEM) images and diameter distribution of nanofiber films at different injection rates. (Spinning process parameters: Collector distance of 17 cm, solution concentration of 10% (*w*/*w*), and spinning voltage of 13 kV. The diagram on the right depicts the distribution of nanofiber diameters).

**Figure 5 polymers-15-03585-f005:**
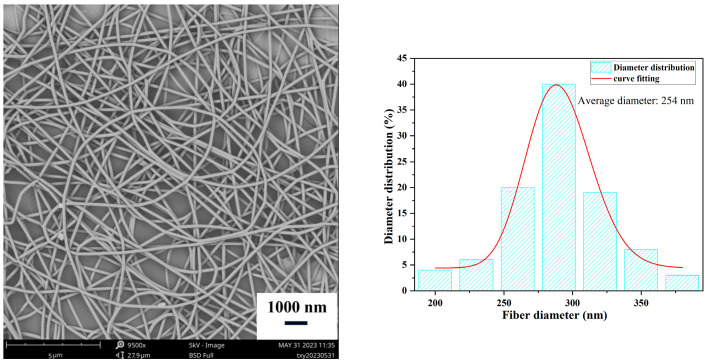
Scanning electron microscope (SEM) images and diameter distribution of the antibacterial nanofiber film under optimal conditions. (Spinning process parameters: Collector distance of 17 cm, solution concentration of 10% (*w*/*w*), spinning voltage of 13 kV, and injection rate of 0.5 mL/h).

**Figure 6 polymers-15-03585-f006:**
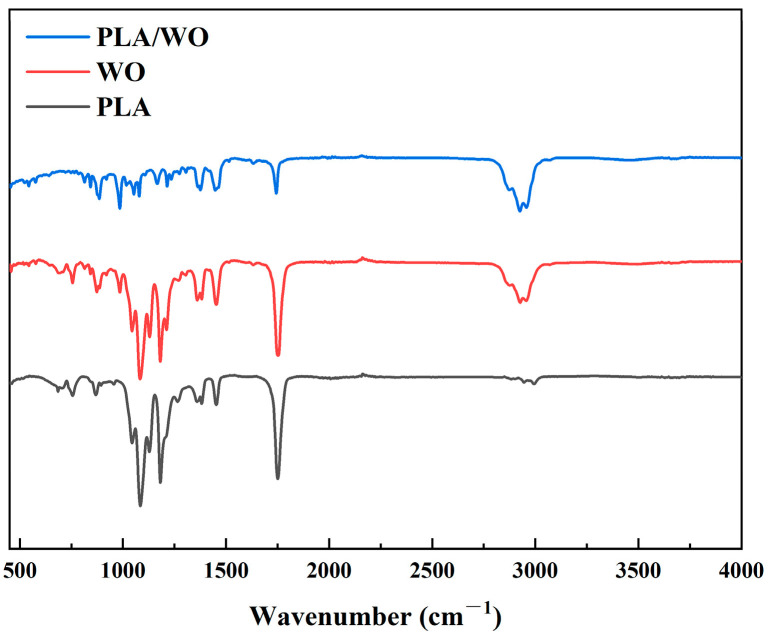
Fourier Transform Infrared (FTIR) spectrum of the nanofiber membrane.

**Figure 7 polymers-15-03585-f007:**
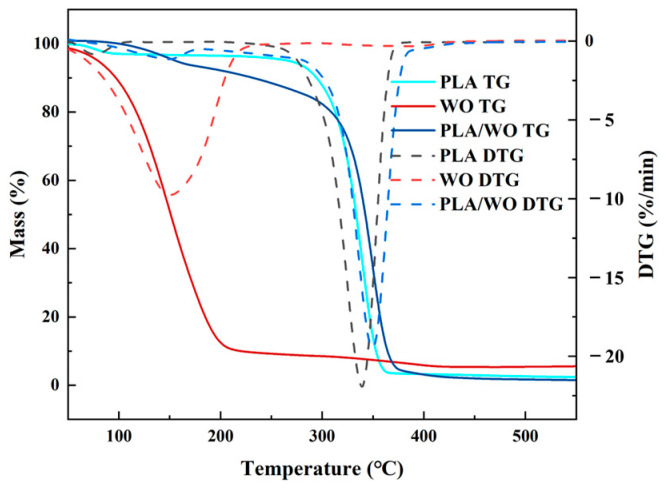
TG curve of the nanofiber membrane.

**Figure 8 polymers-15-03585-f008:**
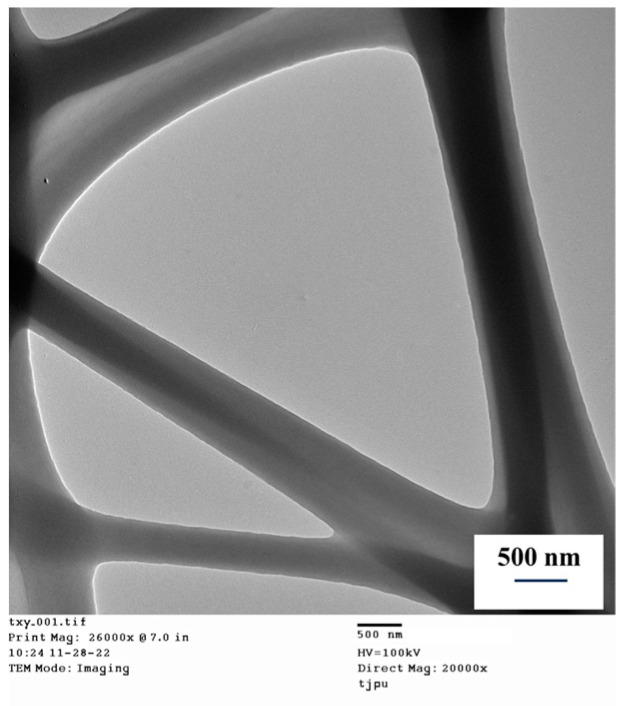
TEM image of the nanofiber membrane.

**Figure 9 polymers-15-03585-f009:**
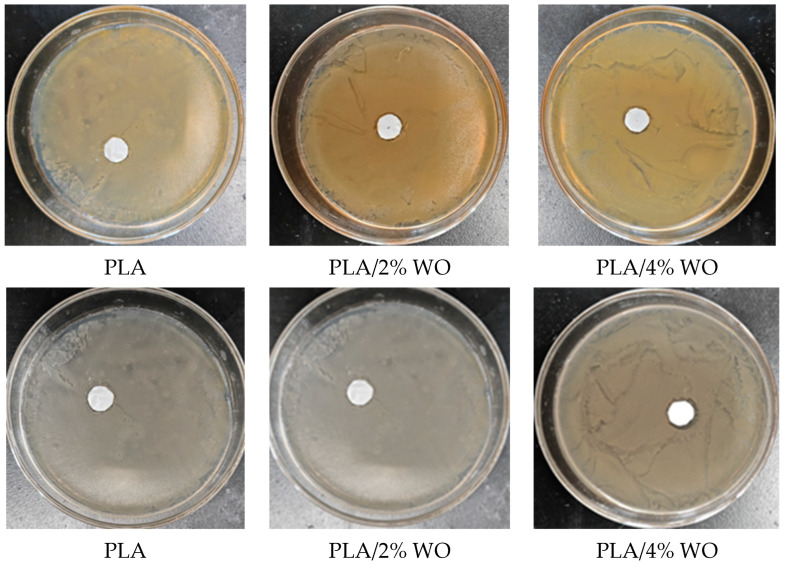
Antimicrobial effects of antibacterial nanofiber membranes with different wormwood oil concentrations against *S. aureus* (**top**) and *E. coli* (**bottom**).

**Table 1 polymers-15-03585-t001:** Viscosity of solutions at different PLA concentrations.

Solution	PLA (%, *w*/*w*)	Viscosity (Pa·s)	Diameter Distribution (nm)	Average Diameter (nm)
PLA	8	0.253	240–450	335
PLA	10	0.361	190–360	260
PLA	12	0.432	240–470	380

**Table 2 polymers-15-03585-t002:** Inhibition Zones of Nanofiber Membranes with Different WO Content.

Microorganism	Film	Inhibition Zone Diameter (mm)
*S. aureus*	PLA	1.0
	PLA/2% WO	2.0
	PLA/4% WO	3.5
*E. coli*	PLA	2.0
	PLA/2% WO	3.5
	PLA/4% WO	4.5

## Data Availability

We have not created new data.

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
