# Peer review of "Preparation and Characterization of a One-Step Electrospun Poly(Lactic Acid)/Wormwood Oil Antibacterial Nanofiber Membrane"

_polymers, 2023, doi:10.3390/polym15173585_

Round 1

Reviewer 1 Report

The authors have described a way of preparing antimicrobial nano fibers by incorporating essential oil into polylactic acid by elecrospinning. There are major revisions that the authors need to do in order to substantiate some of the claims and the research methodology must be improved considerably to make it robust. Please see the comments below

1. Throughout the document the authors have used spinning solution, spinning voltage, spinning distance. The word 'spinning' is used excessively. When the manuscript is about electrospinning it is understood after the first couple of instances that authors refer to electrospinning conditions and therefore 'spinning' before the process parameters can be omitted.

2. Line 26-27- Be specific on what antibacterial effect were observed and specify by numbers. Saying 'good antibacterial effects' is very vague.

3. Line 40- nanofiber based antimicrobial materials are not new. There are numerous articles that in the last two decades have published on antimicrobial incorporated in nanofiber materials.

4. et al. needs to be italicized throughout the document.

5. Line 80- PLA is not new. Researchers in the polymers field have been working with it for long now.

6. Line 88-90- what is an inclusion complex? This sentence needs to be clarified or modified to provide clarity. On line 90- Mention specifically that these are inhibition zone diameters (for increased antibacterial activity). Otherwise the reader doesn't have any context on what is being measured.

7. The names of the bacteria must be italicized throughout the document.

8. Line 99- PLA is not inherently antimicrobial. So enhance is the wrong word. The study aimed to 'impart' antimicrobial properties to PLA......

9. Line 100- Authors mention wormwood essential oil here, in later instances in line 113 they mention mugwort oil, in line 241 it is lavender oil and later on in some instances they mention artemisia essential oil. Authors have to be consistent with that essential oil is being tested and use this terminology consistently throughout the manuscript.

10. Line 129- In a technical paper never refer to it as golden Staph. Please use Staphylococcus aureus. Mention the sources for both S. aureus and E. coli (for e.g. ATCC #)

11. Line 132- Serial dilution.

12. Line 133- need to elaborate what PCL stands for before abbreviating.

13. All the SEM images in Figures 1-4 are very small and not legible. It is advisable to enlarge the SEM images and have the fiber diameter distribution under the SEM images for better clarity.

14. Figure 6- Assign peaks with the corresponding vibrations for clarity. Why is the peak at 2960 cm-1 decreasing for PLA/WO while it is clearly present for each of the individual components?

15. Line 283- shouldnt these values be reversed? starts at 102 C and maximum rate occurs at 105 C?

16- Line 293- The authors cannot say this for sure without corroborating evidence/elemental mapping. It could just be that the fibers are dense in the middle and they stretch out at the edges. EDS or ToF-SIMS mapping of the element will be essential to make this definitive statement. 

17. Line 304-307, 320-322- From the photos, there is no evidence of an increase in the inhibition zone. Except for PLA/4% WO it is hard to discern from the figures where is the edge of the inhibition zone. The authors must get close up photos of the inhibition zones along with the zone diameters measurements overlayed on the photos to say definitely if the inhibition zone diameter is increasing with increase in %WO. Moreover, these experiments must be performed with samples in triplicate in the least.

Please look at the attached file for comments and edits.

Author Response

  1. Throughout the document the authors have used spinning solution, spinning voltage, spinning distance. The word 'spinning' is used excessively. When the manuscript is about electrospinning it is understood after the first couple of instances that authors refer to electrospinning conditions and therefore 'spinning' before the process parameters can be omitted.

Thank you for your comment. The manuscript has been revised accordingly.

  1. Line 26-27- Be specific on what antibacterial effect were observed and specify by numbers. Saying 'good antibacterial effects' is very vague.

Thank you for your comment .In the manuscript I have clearly written about the changes in the circle of inhibition.

  1. Line 40- nanofiber based antimicrobial materials are not new. There are numerous articles that in the last two decades have published on antimicrobial incorporated in nanofiber materials.

Thank you for your comment. The manuscript has been revised accordingly. This study aimed to impart antibacterial properties of polylactic acid (PLA) polymer by incorporating wormwood essential oil using a one-step electrospinning method.

  1. et al. needs to be italicized throughout the document.

Thank you for your comment. The manuscript has been revised accordingly.

  1. Line 80- PLA is not new. Researchers in the polymers field have been working with it for long now.

Thank you for your comment. The manuscript has been revised accordingly.

  1. Line 88-90- what is an inclusion complex? This sentence needs to be clarified or modified to provide clarity. On line 90- Mention specifically that these are inhibition zone diameters (for increased antibacterial activity). Otherwise the reader doesn't have any context on what is being measured.

Thank you for your comment. The manuscript has been revised accordingly. The results indicated that due to the inclusion complexation of β-CD with TR, the degradation of TR in the environment was reduced, resulting in significantly larger inhibition zone diameters of the PLA/TR/β-CD nanofiber membrane against Escherichia coli (E.coli)and Staphylococcus aureus (S.aureus) compared to the PLA/TR nanofiber membrane (3.7 cm VS 3.0 cm; 3.50 cm VS 2.83 cm).

  1. The names of the bacteria must be italicized throughout the document.

Thank you for your comment. The manuscript has been revised accordingly.

  1. Line 99- PLA is not inherently antimicrobial. So enhance is the wrong word. The study aimed to 'impart' antimicrobial properties to PLA......

Thank you for your comment. The manuscript has been revised accordingly.

  1. Line 100- Authors mention wormwood essential oil here, in later instances in line 113 they mention mugwort oil, in line 241 it is lavender oil and later on in some instances they mention artemisia essential oil. Authors have to be consistent with that essential oil is being tested and use this terminology consistently throughout the manuscript.

Thank you for your comment. This was an error in my writing, and all that was used in the text was wormwood oil.

  1. Line 129- In a technical paper never refer to it as golden Staph. Please use Staphylococcus aureus. Mention the sources for both  aureusand E. coli (for e.g. ATCC #)

Thank you for your comment. The manuscript has been revised accordingly.

  1. Line 132- Serial dilution.

Thank you for your comment. The manuscript has been revised accordingly.

  1. Line 133- need to elaborate what PCL stands for before abbreviating.

Thank you for your comment. That was a mistake in my writing, it should be PLA

  1. All the SEM images in Figures 1-4 are very small and not legible. It is advisable to enlarge the SEM images and have the fiber diameter distribution under the SEM images for better clarity.

Thank you for your comment. I've separated the SEM images from the diameter distribution map.

  1. Figure 6- Assign peaks with the corresponding vibrations for clarity. Why is the peak at 2960 cm-1decreasing for PLA/WO while it is clearly present for each of the individual components?

Thank you for your comment. I rechecked the IR spectra and the first curve should be an IR spectrogram of PLA/WO.

  1. Line 283- shouldnt these values be reversed? starts at 102 C and maximum rate occurs at 105 C?

Thank you for your comment. This was an error in my writing and has been corrected.

16- Line 293- The authors cannot say this for sure without corroborating evidence/elemental mapping. It could just be that the fibers are dense in the middle and they stretch out at the edges. EDS or ToF-SIMS mapping of the element will be essential to make this definitive statement. 

Thank you for your comment. This is just a preliminary judgment. In later experiments, I will perform a variety of verifications.

  1. Line 304-307, 320-322- From the photos, there is no evidence of an increase in the inhibition zone. Except for PLA/4% WO it is hard to discern from the figures where is the edge of the inhibition zone. The authors must get close up photos of the inhibition zones along with the zone diameters measurements overlayed on the photos to say definitely if the inhibition zone diameter is increasing with increase in %WO. Moreover, these experiments must be performed with samples in triplicate in the least.

Thank you for your comment. I have written the measured size of the circle of inhibition in a new table. The experiment was conducted 3 times, and the experimental pattern was the same in each case. I chose one of the trials to show.

Reviewer 2 Report

The reviewed manuscript entitled “Preparation and Characterization of One-Step Electrospun Poly (lactic acid)/Artemisia Essential Oil Antibacterial” aims to improve the antibacterial properties of the polylactic-acid (PLA) polymer by incorporating wormwood essential oil through a one-step electrospinning method. However, the article is currently in a poorly written state and has been submitted without undergoing necessary revisions. To enhance the manuscript's readability and clarity, several major concerns must be addressed before further processing can take place:

1-     The motivation for the study and the research gap are not clear enough. Please demonstrate in the introduction of the paper, the novelty of this research in relation to other thematically similar research papers such as “LIU Yang, NIU Jianxing, LI Yuyao, QIAN Xiaoming, WANG Liang, LIU Yong. Preparation Artemisia/polyacrylonitrile nanofiber composites and their super-hydrophilic and long-term antibacterial properties [J]. Acta Materiae Compositae Sinica, 2022, 39(5): 2258-2268. doi: 0.13801/j.cnki.fhclxb.20210701.003  .

2-     In the abstract: “….the formed antibacterial nanofiber membrane had a smooth morphology without bead formation, with an average diameter of 260.5 nm”, while in the introduction: “By optimizing the electrospinning conditions, nanocomposite materials with smooth surface, an average diameter of 300 nanometers, and excellent antibacterial properties were successfully prepared.”. Please unify the average diameter of your prepared nanofibers.

3-     L81: “…..as well as high tensile strength and elongation [16-19].” Avoid lumped references; a short comment should be included for each reference or two references in the same subject.

4-     L241: “Therefore, for a solution composition of 10% (w/w) PCL/4% (w/w) lavender essential oil”. What is PCL? Please define any abbreviations for the first time appear before using in the text. Moreover, what was used exactly in the study, Artemisia or lavender essential oil? What is the chemical composition of the used oil? Please characterize the exact used oil via a suitable test.

5-     Figure 5, please provide the original SEM image with the original scale bar and other printed information. Original pictures of experimentation add to the novelty and quality of the paper.

6-     L266: “The peaks at 3460 cm-1 and …..”. There is no peak at 3460 cm-1, please clarify.

7-     Figure 6, Fourier Transform Infrared: Why the clear peaks at 2960 cm-1 for both PLA and WO were almost disappeared when mixing the two component together? Discussion is lack of scientific explanation for the obtained results. Authors should attribute the results achieved to a clear scientific reason.

8-     Figure 7, TG curve: Very strange behavior was noticed for the PLA/WO TG composite. Why does more than 50% from the composite were degraded at a very low temperatures slightly above 100, while the included percentage of oil does not exceed 6% at its maximum? What happened to the PLA that was supposed not to start degradation before 300 degrees? It seems that something is wrong .. Please give your scientific explanation. Further investigation and analysis are required to determine the exact cause of this unusual degradation pattern.

9-     Figure 8, TEM: “From Figure 8, it can be observed that the antibacterial nanofiber membrane consists of two components. The inner component is WO, while the outer component is PCL. WO is encapsulated within PLA, forming a core-shell structure of PLA and WO.” However, there is currently no any evidence for the claimed structure. The current TEM image is not showing any core-shell structure, which is impossible to be generated without using a core-shell needle structure. Therefore, TEM image is useless and can be omitted.

10- In Figure 9, the inhibition diameter is currently unclear and cannot be accurately measured. To address this issue, it is recommended to zoom in while capturing the images. By doing so, the different inhibition zones will become more visible and comparable.

11- Conclusion needs to be reconstructed focusing on the study findings.

12- The English language used in the paper is to be revised and improved before the subsequent manuscript submission. Please, read the text carefully before the next submission of the paper.

Author Response

1-The motivation for the study and the research gap are not clear enough. Please demonstrate in the introduction of the paper, the novelty of this research in relation to other thematically similar research papers such as “LIU Yang, NIU Jianxing, LI Yuyao, QIAN Xiaoming, WANG Liang, LIU Yong. Preparation Artemisia/polyacrylonitrile nanofiber composites and their super-hydrophilic and long-term antibacterial properties [J]. Acta Materiae Compositae Sinica, 2022, 39(5): 2258-2268. doi: 0.13801/j.cnki.fhclxb.20210701.003  .

Thank you for your comment. The antimicrobial substance used by LIU Yang is wormwood powder, and the antimicrobial substance used in this article is wormwood oil. Theoretically, wormwood oil has a better antibacterial effect.

2-In the abstract: “….the formed antibacterial nanofiber membrane had a smooth morphology without bead formation, with an average diameter of 260.5 nm”, while in the introduction: “By optimizing the electrospinning conditions, nanocomposite materials with smooth surface, an average diameter of 300 nanometers, and excellent antibacterial properties were successfully prepared.”. Please unify the average diameter of your prepared nanofibers.

Thank you for your comment. This was an error in my writing and have changed 300 to 260.5.

3-L81: “…..as well as high tensile strength and elongation [16-19].” Avoid lumped references; a short comment should be included for each reference or two references in the same subject.

Thank you for your comment. The manuscript has been revised accordingly.

4-L241: “Therefore, for a solution composition of 10% (w/w) PCL/4% (w/w) lavender essential oil”. What is PCL? Please define any abbreviations for the first time appear before using in the text. Moreover, what was used exactly in the study, Artemisia or lavender essential oil? What is the chemical composition of the used oil? Please characterize the exact used oil via a suitable test.

Thank you for your comment. The materials used for the test were PLA and wormwood oil. According to the literature and our laboratory tests, the main components of wormwood essential oil are camphor-4 and α-terpineol. Wormwood essential oil has excellent antibacterial properties.

5-Figure 5, please provide the original SEM image with the original scale bar and other printed information. Original pictures of experimentation add to the novelty and quality of the paper.

Thank you for your comment. The manuscript has been revised accordingly.

6-L266: “The peaks at 3460 cm-1 and …..”. There is no peak at 3460 cm-1, please clarify.

Thank you for your comment. This was an error in writing and I have made changes.

7-Figure 6, Fourier Transform Infrared: Why the clear peaks at 2960 cm-1for both PLA and WO were almost disappeared when mixing the two component together? Discussion is lack of scientific explanation for the obtained results. Authors should attribute the results achieved to a clear scientific reason.

Thank you for your comment. I rechecked the IR spectra and the first curve should be an IR spectrogram of PLA/WO.

8-Figure 7, TG curve: Very strange behavior was noticed for the PLA/WO TG composite. Why does more than 50% from the composite were degraded at a very low temperatures slightly above 100, while the included percentage of oil does not exceed 6% at its maximum? What happened to the PLA that was supposed not to start degradation before 300 degrees? It seems that something is wrong .. Please give your scientific explanation. Further investigation and analysis are required to determine the exact cause of this unusual degradation pattern.

Thank you for your comment. This is indeed in error. I have re-run the test, please check it out.

9-Figure 8, TEM: “From Figure 8, it can be observed that the antibacterial nanofiber membrane consists of two components. The inner component is WO, while the outer component is PCL. WO is encapsulated within PLA, forming a core-shell structure of PLA and WO.” However, there is currently no any evidence for the claimed structure. The current TEM image is not showing any core-shell structure, which is impossible to be generated without using a core-shell needle structure. Therefore, TEM image is useless and can be omitted.

Thank you for your comment. This is just a preliminary judgment. In later experiments, I will perform a variety of verifications.

10-In Figure 9, the inhibition diameter is currently unclear and cannot be accurately measured. To address this issue, it is recommended to zoom in while capturing the images. By doing so, the different inhibition zones will become more visible and comparable.

Thank you for your comment. I have written the measured size of the circle of inhibition in a new table.

11-Conclusion needs to be reconstructed focusing on the study findings.

Thank you for your comment. I've rewritten the conclusion.

12- The English language used in the paper is to be revised and improved before the subsequent manuscript submission. Please, read the text carefully before the next submission of the paper.

Thank you for your comment. I have re-read the whole article and made changes.

Round 2

Reviewer 1 Report

The author has made the necessary changes and reformatted some of the issues pointed out with the manuscript. However, some minor issues are present and if corrected the manuscript can be accepted.

1. The authors still have not mentioned the sources of the bacterial strains. "They were provided by the laboratory" is not enough. Identification of the strain numbers if commercially produced is necessary. If they are isolated by their laboratory from a certain location, the source from which it is isolated must be mentioned. If someone wants to repeat the experiments the authors performed and reproduce the same results, they will need to know all the specific information.

2. In the "3.4. Transmission electron microscopy section", the authors have confirmed the encapsulation of WO in PLA fibers and these are definitive statements. However, in the responses the author says this is a preliminary judgment and they will perform more studies to verify it. Either verify it and then make the conclusive statement or remove the confirmatory statement and say that encapsulation is just a hypothesis and further studies must be conducted to confirm it. 

3. The author still needs to mention explicitly in the text in the "Antibacterial Performance and Analysis" section that the samples were tested in triplicate.

Author Response

  1. The authors still have not mentioned the sources of the bacterial strains. "They were provided by the laboratory" is not enough. Identification of the strain numbers if commercially produced is necessary. If they are isolated by their laboratory from a certain location, the source from which it is isolated must be mentioned. If someone wants to repeat the experiments the authors performed and reproduce the same results, they will need to know all the specific information.

Thank you for your comment. The manuscript has indicated the source and identification numbers of bacterial strains.

  1. In the "3.4. Transmission electron microscopy section", the authors have confirmed the encapsulation of WO in PLA fibers and these are definitive statements. However, in the responses the author says this is a preliminary judgment and they will perform more studies to verify it. Either verify it and then make the conclusive statement or remove the confirmatory statement and say that encapsulation is just a hypothesis and further studies must be conducted to confirm it. 

Thank you for your comment. The manuscript has removed the confirmatory statements.

  1. The author still needs to mention explicitly in the text in the "Antibacterial Performance and Analysis" section that the samples were tested in triplicate.

Thank you for your comment. The manuscript explicitly mentions that the samples underwent three replicate antimicrobial tests.

Reviewer 2 Report

The revised version of the manuscript is not satisfactory, as the authors have not provided amendments to many of the suggested queries. The doubts of this reviewer persist regarding the following points:

8.a- The newly added TGA results for PLA/WO are questionable and raise doubts. The sharp jump in temperature (from 142 to 307℃) without any change in weight loss needs to be scientifically explained. At a heating rate of 10℃/min, this jump took about 16.5 minutes with no loss in composite weight, which contradicts the TGA results of the separate composite components, PLA and WO. Furthermore, it is necessary to clarify how the TGA results changed without affecting the DTG graph. Please support your claim and explanation with relevant references.

8.b- In the updated TGA results for PLA/WO, the final residuals were less than those of PLA and WO separately, which is illogical and requires a scientific explanation as well as supportive references.

8.c- Line 312: "..... materials heated from 30℃ to 600℃ at a heating rate of 10℃/min." Please adjust the full range scale of the graph or correct the text accordingly.

9- L328: “From Figure 8, it can be observed that the antibacterial nanofiber membrane consists of two components. The inner component is WO, while the outer component is PLA. WO is encapsulated within PLA, and its dispersion in PLA is random, forming a core-shell structure as well as mixed structures”. The authors still claim that the produced membrane forms a core-shell structure without providing any evidence. The use of TEM is deemed ineffective and does not contribute to the manuscript. Their response stating, "This is just a preliminary judgment. In later experiments, I will perform a variety of verifications," is unsatisfactory and cannot be accepted.

10- It seems that there is a discrepancy between the information provided in the experimental part and the data presented in table 2. The experimental part states in L158 that the sample diameter ranges from 6 mm to 1 cm, while table 2 shows a maximum inhibition zone diameter of 1 cm. Based on this information, it appears that there may be cases where the sample diameter is equal to or smaller than the maximum inhibition zone diameter, resulting in no real inhibition. Additionally, for a fair comparison, it is recommended to use a fixed sample diameter.

11- It is evident that the current version of the conclusion lacks clarity and coherence. The excessive emphasis on technical details, such as FTIR and its peaks, has overshadowed the quantitative findings of the study. Furthermore, the statement made in line 374, claiming that "TEM images clearly demonstrated the successful combination of WO and PLA," is not accurate and has not been verified yet.

Author Response

8.a- The newly added TGA results for PLA/WO are questionable and raise doubts. The sharp jump in temperature (from 142 to 307℃) without any change in weight loss needs to be scientifically explained. At a heating rate of 10℃/min, this jump took about 16.5 minutes with no loss in composite weight, which contradicts the TGA results of the separate composite components, PLA and WO. Furthermore, it is necessary to clarify how the TGA results changed without affecting the DTG graph. Please support your claim and explanation with relevant references.

Thank you for your comment. I have conducted rigorous testing on the samples again, and the TG curve of PLA/WO has been revised. Please review it again.

8.b- In the updated TGA results for PLA/WO, the final residuals were less than those of PLA and WO separately, which is illogical and requires a scientific explanation as well as supportive references.

Thank you for your comment. WO is prone to volatilization and degradation, with almost complete volatilization and degradation of WO. The residue in PLA/WO consists mainly of PLA thermal degradation residue, therefore its degradation residue rate is similar to that of PLA.

8.c- Line 312: "..... materials heated from 30℃ to 600℃ at a heating rate of 10℃/min." Please adjust the full range scale of the graph or correct the text accordingly.

Thank you for your comment. The text in the manuscript has been corrected.

9- L328: “From Figure 8, it can be observed that the antibacterial nanofiber membrane consists of two components. The inner component is WO, while the outer component is PLA. WO is encapsulated within PLA, and its dispersion in PLA is random, forming a core-shell structure as well as mixed structures”. The authors still claim that the produced membrane forms a core-shell structure without providing any evidence. The use of TEM is deemed ineffective and does not contribute to the manuscript. Their response stating, "This is just a preliminary judgment. In later experiments, I will perform a variety of verifications," is unsatisfactory and cannot be accepted.

Thank you for your comment. The manuscript has removed the confirmatory statements.

10- It seems that there is a discrepancy between the information provided in the experimental part and the data presented in table 2. The experimental part states in L158 that the sample diameter ranges from 6 mm to 1 cm, while table 2 shows a maximum inhibition zone diameter of 1 cm. Based on this information, it appears that there may be cases where the sample diameter is equal to or smaller than the maximum inhibition zone diameter, resulting in no real inhibition. Additionally, for a fair comparison, it is recommended to use a fixed sample diameter.

Thank you for your comment. The expressions in the manuscript contained errors, and I have made the necessary corrections.

11- It is evident that the current version of the conclusion lacks clarity and coherence. The excessive emphasis on technical details, such as FTIR and its peaks, has overshadowed the quantitative findings of the study. Furthermore, the statement made in line 374, claiming that "TEM images clearly demonstrated the successful combination of WO and PLA," is not accurate and has not been verified yet.

Thank you for your comment. I have reconstructed the conclusion, removing the excessive emphasis on infrared spectroscopy and the deterministic statements in TEM. Kindly review it again. 

Round 3

Reviewer 2 Report

Can be published